# Optimal Bowel Preparation Method to Visualize the Distal Ileum via Small Bowel Capsule Endoscopy

**DOI:** 10.3390/diagnostics13203269

**Published:** 2023-10-20

**Authors:** Daisuke Kametaka, Mamoru Ito, Seiji Kawano, Shuhei Ishiyama, Akiko Fujiwara, Junichirou Nasu, Masao Yoshioka, Junji Shiode, Kazuhide Yamamoto, Masaya Iwamuro, Yoshiro Kawahara, Hiroyuki Okada, Motoyuki Otsuka

**Affiliations:** 1Department of Gastroenterology and Hepatology, Okayama University Graduate School of Medicine, Dentistry, and Pharmaceutical Sciences, Okayama 700-8558, Japan; johnrambo851@yahoo.co.jp (D.K.); piuz1nfw@s.okayama-u.ac.jp (S.K.); pr145h2k@s.okayama-u.ac.jp (M.I.); yoshirok@md.okayama-u.ac.jp (Y.K.); otsukamoto@okayama-u.ac.jp (M.O.); 2Department of Internal Medicine, Okayama Saiseikai General Hospital, Okayama 700-8511, Japan; shuhei.ishiyama@gmail.com (S.I.); furst2002@gmail.com (A.F.); jnasu@saiseidr.jp (J.N.); masanga114@gmail.com (M.Y.); siodeju@gmail.com (J.S.); kazuhide@okayamasaiseikai.or.jp (K.Y.); 3Department of Internal Medicine, Japanese Red Cross Himeji Hospital, Himeji 670-8540, Japan; hiro@md.okayama-u.ac.jp

**Keywords:** distal ileum, visual quality, scoring system, timing of ingestion, small intestine

## Abstract

Small bowel capsule endoscopy (SBCE) is a convenient and minimally invasive method widely used to evaluate the small intestine. However, especially in the distal ileum, visualization of the intestinal mucosa is frequently hampered by the remaining intestinal contents, making it difficult to detect critical lesions. Although several studies have reported on the efficacy of bowel preparation before SBCE, no standardized protocol has been established. Herein, we determined the optimal preparation method for better visualization of the distal ileum using SBCE. We retrospectively analyzed 259 consecutive patients who had undergone SBCE between July 2009 and December 2019, divided into three groups: Group A (no preparation except overnight fasting), Group B (ingestion of 1–2 L polyethylene glycol 4 h before colonoscopy after overnight fasting and performing SBCE immediately after colonoscopy), and Group C (ingestion of 0.9 L magnesium citrate [MC] before SBCE after overnight fasting). The visibility of the intestinal mucosa in the first 10 min and at the last 10 min during the period of observation of the distal ileum was examined using a scoring system and compared. The visibility of the images captured by SBCE was assessed based on the scoring of the degree of bile/chyme staining, residual fluid and debris, brightness, bubble reduction, and visualized mucosa. The status of intestinal collapse was also assessed. In the first 10 min of observation of the distal ileum, no significant differences were detected among the groups. In the last 10 min, significantly better images were acquired in Group C in terms of bile/chyme staining, brightness, bubble reduction, and visualized mucosa. Bowel preparation using a low-dose MC solution 2 h before SBCE provided significantly higher-quality images of the distal ileum. Further optimization, such as the timing of initiating the preparation, is necessary to determine the optimal regimen for bowel preparation prior to SBCE.

## 1. Introduction

Owing to its anatomical features, the small intestine is clinically difficult to evaluate using endoscopy. Small bowel capsule endoscopy (SBCE) is widely used as an alternative method to evaluate various small intestinal lesions with minimal invasiveness. However, visualization of the intestinal mucosa using SBCE is frequently hampered by residual intestinal contents (e.g., food residue, air bubbles, and unclear turbid intraluminal fluid), especially in the distal small intestine, where pathological findings are clinically more frequent.

To obtain better-visualized images of the intestinal mucosa by SBCE, several bowel preparation methods before SBCE tests have been examined, and their effectiveness in obtaining better-visualized intestinal images has been reported [1,2,3,4,5]. A Dutch study evaluated 90 patients who were randomized to either undergo regular preparation consisting of an after-clear liquid diet and overnight fast before SBCE or receive 1 or 2 L polyethylene glycol (PEG) solution before SBCE. In this study, good visualization of the terminal ileum was achieved in only 25% of the patients after standard preparation, while after preparation with both 1 and 2 L of PEG solution, mucosal visualization improved to 52% and 72% of patients, respectively [2]. Wi JH et al. reported the efficacy of bowel preparation with two doses of 45 mL of sodium phosphate with water in the afternoon and evening of the day before the procedure, followed by at least 2 L of water [3]. Nouda et al. stated that the combined use of 1L of PEG and dimethylpolysiloxane 3 h before the examination was effective [4]. Esaki et al. reported the effectivity of receiving 0.9 L of magnesium citrate (MC) 3 h before the examination [5]. Additionally, in the recent report by Choi CW et al., the timing of bowel preparation within 6 h between the last ingestion of PEG and swallowing of SBCE improved the quality of small bowel visibility [6]. However, the efficacy of bowel preparation prior to SBCE remains controversial. The main reason for this controversy is that although bowel preparation increases visibility by eliminating the intestinal contents, leading to the improvement of diagnostic efficacy, taking a certain amount of laxative for such preparation may reduce the merits of SBCE, such as minimal invasiveness and also reduce the patient’s tolerability. Hansel S et al. reported that the combined use of PEG solution, simethicone, and metoclopramide prior to SBCE did not improve small bowel visualization but did significantly increase patient discomfort [7]. Hookey L et al. found no benefit in overall or distal small bowel visualization with active preparation using either PEG or sodium picosulfate plus magnesium sulfate compared with clear fluids only [8]. Lamba M et al. reported that the use of a PEG preparation before SBCE did not result in improved diagnostic yield or small bowel visualization quality [9]. In this current situation, the European Society of Gastrointestinal Endoscopy published a Guideline Update in 2022 and stated that adequate SB visualization is a crucial element in ensuring a reliable assessment of the small intestine. On the other hand, they also stated that the evidence recommending the use of purgative solutions prior to SBCE remains somewhat controversial [10]. Therefore, no standardized global protocol for bowel preparation preceding SBCE has been established. Deeper into the ileum, the visuality of SBCE images tends to worsen because dark intestinal fluid and/or air bubbles have increased. However, the distal ileum, including the terminal ileum, is one of the sites with abnormal findings in several small intestinal diseases.

Herein, by comparing three preparation methods, we aimed to determine the optimal intestinal preparation method before SBCE by scoring the visibility of the intestinal mucosa at the distal ileum, where visibility is often hampered by intestinal contents.

## 2. Materials and Methods

### 2.1. Patient and Ethics

We enrolled 355 consecutive patients (male to female ratio: 214:141) who had undergone SBCE between July 2009 and December 2019 at Okayama Saiseikai General Hospital and retrospectively assessed their clinical records. Participants were divided into three groups according to the preparation methods used. Group A: conventional overnight fasting without specific intestinal preparation methods (no preparation). Group B: Ingestion of 1–2 L of PEG solution approximately 4 h before examination after overnight fasting. Ingestion was performed until no residual intestinal contents were observed. SBCE was performed immediately after the colonoscopy (PEG). Group C: ingestion of 0.9 L of MC solution 2 h before the SBCE after overnight fasting (MC). When reimbursement for small bowel CE was first approved by the national medical insurance system in Japan in 2007, its indication was restricted to OGIB; that is, gastrointestinal bleeding of unknown origin that persists or recurs after negative results of upper endoscopy and colonoscopy, thereafter the indications for CE were expanded to ‘patients known to have or suspected of having small bowel disease’ in 2012 [11]. Therefore, at least until July 2012, it was necessary to confirm that there was no cause by upper endoscopy and colonoscopy before performing SBCE. Considering that small intestinal bleeding is less frequent among gastrointestinal bleeding, upper endoscopy and colonoscopy are still often performed before SBCE, especially in the case of OGIB, even after a short period of time has passed since the insurance coverage was expanded. Hence, at our institution, colonoscopy is frequently performed before SBCE, and because PEG is usually used for bowel preparation before colonoscopy, we set Group B as a group using PEG for colonoscopy and SBCE examination on the same day. No prokinetics were used to prepare the three groups. This study was approved by the Ethical Review Board of Okayama Saiseikai General Hospital (No. 180205).

### 2.2. SBCE Examination

SBCE was performed using a PillCam SB2 or SB3 (Given Imaging Ltd., Yokneam, Israel). The images were analyzed with RAPID Reader 6.5 or 8 software on a RAPID workstation (software and workstation from Given Imaging Ltd.). For each SBCE test, we defined the time from the beginning of the duodenum to the arrival of the cecum as the “total small bowel transit time.” This “total small bowel transit time” was divided into four equal parts, and the last quarter of the time was defined as the period observing the distal ileum. To evaluate the visibility of the intestinal mucosa at the distal ileum in detail, the first 10 min and last 10 min during the observation period of the distal ileum were selected for the retrospective assessment of the visibility of the SBCE images.

### 2.3. Assessment of the SBCE Images and Scoring System

SBCE images were reviewed by two expert gastroenterologists (D.K and M.I), who were blinded to the preparation methods used. Calibration of the scoring system to ensure standardization was performed by another expert gastroenterologist (S.K). The visibility of the SBCE images was assessed on six elements revised by the validated quantitative assessment score described by Brotz et al. [12]: (1) bile/chyme staining, (2) remaining fluid and debris, (3) brightness, (4) bubble reduction, (5) clarity of the visualized mucosa, and (6) existence of intestinal collapse. The first three elements were evaluated using a four-point scale (Figure 1 and Table 1). Bubble reduction and the clarity of the visualized mucosa were initially evaluated as percentages and then scored with a four-point scale, classified into <20%, 20–49%, 50–79%, and ≥80% (Figure 1 and Table 1). Intestinal collapse was defined as “observed (collapsed)” when the internal lumen was visible only with less than one-fifth of the screen and difficult to assess the intestinal mucosa (Figure 1 and Table 1). When each score did not match between the two gastroenterologists, the final scoring was made after discussion.

### 2.4. Statistical Analysis

All statistical analyses were performed using JMP Pro 15 (SAS Institute Inc., Cary, NC, USA). Data were expressed as the mean ± standard deviation. Categorical variables were tested using the chi-squared test or Fisher’s exact test, as appropriate. Nonparametric data were evaluated using the Kruskal–Wallis test. Subsequently, the Wilcoxon rank-sum test was used to compare the groups. *p* values < 0.05 were considered statistically significant.

## 3. Results

### 3.1. Patient Characteristics and Study Overview

Among 355 consecutive patients who had undergone SBCE between July 2009 and December 2019, 96 were excluded because of massive bleeding (*n* = 18), unreached cecum (*n* = 16), recording failure (*n* = 3), and unknown preparation used (*n* = 4). Additionally, cases receiving 0.18 L of MC solution before overnight fasting (*n* = 24) and those receiving 1–2 L of PEG solution after overnight fasting without colonoscopy (*n* = 31) were excluded because the former regimen is currently uncommon, and the PEG dose was not clearly recorded. In total, 259 patients were enrolled in this study. Depending on the bowel preparation methods used before SBCE, 51 patients were classified into Group A (No preparation), 123 into Group B (PEG), and 85 into Group C (MC) (Figure 2 and Table 2). The examination of the small intestine for OGIB was the most frequent indication for SBCE (Table 2). No significant differences in age, sex, gastric transit time, small bowel transit time, or indication for SBCE were observed among these three groups (Table 2).

### 3.2. Differences in Visibility at the Last 10 min during the Period Observing the Distal Ileum

The visibility of the SBCE images during the first 10 min of observation of the distal ileum was not significantly different among the groups (Figure 3). Intestinal collapse was also not significantly different between groups (Figure 4). However, the visibility of the distal ileum was significantly different in the last 10 min of the observation period. During this period, Group C revealed significantly better visibility in terms of bile/chyme staining, brightness, bubbles reduction, and visualized mucosa (Figure 5). Despite no significant difference in fluid and debris, this score tended to be higher in Group C (Figure 4). Regarding intestinal collapse, although no significant differences were observed among the groups, even during the last 10 min, it tended to be more pronounced in Group B (Figure 4).

These results suggest that the ingestion of 0.9 L MC before the examination provides better visibility, especially in the last 10 min, when observing the distal ileum.

## 4. Discussion

Although the clinical demand for SBCE is increasing, no consensus regimen for bowel preparation methods for SBCE has been established. In this study, to determine the best bowel preparation method before SBCE, we compared three different preparation methods. We observed the superiority of ingestion of 0.9 L of MC before SBCE after overnight fasting with regard to the visibility of the small intestine, especially the distal ileum.

SBCE has been widely accepted as a convenient diagnostic tool that enables the minimally invasive observation of small intestinal lesions [13]. After a few upgrades from the original capsule, the resolution and frame rates have improved, and owing to the improvement in battery life, its utility in clinical practice has increased and the indications for SBCE are evolving [11]. However, the mucosal visualization of SBCE is frequently hampered by residual intestinal contents, especially when visualizing the distal ileum. To overcome this poor visibility, several bowel preparation methods have been tested, such as the oral intake of 1 L PEG in the evening the day before examination, the oral intake of a combination of 1 L PEG in the evening the day before examination and of dimethylpolysiloxane 3 h before examination [2,4], and the oral intake of 0.9 L MC 3 h before examination [5]. Although these preparation methods have been reported to be successful, several studies have reported that bowel preparation for SBCE does not improve mucosal visibility [7,8,9,14,15,16,17]. Therefore, no consensus has been established on a regimen for bowel preparation in SBCE. The distal ileum, including the terminal ileum, is a site with abnormal findings in several small intestinal diseases. Therefore, we considered preparation for SBCE important for improving the visibility of the distal ileum. In this study, we focused on the last 10 min of observation of the distal ileum.

We tested different kinds of bowel preparation methods by dividing the cases subjected to SBCE into three groups, depending on the bowel preparation regimens used: Group A, Group B, and Group C. Because at our institution, we commonly perform SBCE immediately after colonoscopy when no abnormality is found by colonoscopy, and because PEG is usually used for bowel preparation before colonoscopy, we set Group B as a group using PEG for colonoscopy and SBCE examination on the same day. Although preparations containing magnesium may impair the visibility of SBCE by contracting the gallbladder through the secretion of cholecystokinin and increasing bile and air bubbles [18,19], some studies have reported the efficacy of MC [5]. Therefore, we selected the use of MC (Group C) as another preparation method for SBCE.

Before this study, we had anticipated that Group B might reveal better visibility of the small intestine because it received sufficient amounts of PEG to clean the intestine to perform colonoscopy, which should theoretically clean the small bowel. However, Group C showed the best visualization of the distal ileum. One possible reason for this unexpected result is that large amounts of PEG may be associated with a more turbid liquid, which interferes with the visibility of the distal small bowel [20]. Another possible reason is the difference in preparation timing. Several studies have shown that taking purgative solutions just before SBCE is useful [4,5,6,20]. Moreover, because taking purgative solutions after the confirmation of the reach of the capsule to the small bowel is useful [21,22], the shorter time required for the preparation in Group C (2 h) than that required in Group B (at least 4 h) might have worked as effectively for the visibility of the distal ileum. In group B, ileal intubation with the injection of water and suction in order to visualize the mucosa may have influenced the results, such as fluids and bubbles. However, in Group B, SBCE had reached the terminal ileum after 5 hours on average. Taking purgative solutions after the confirmation of the reach of the capsule to the small bowel is considered useful [21,22]. Therefore, we consider that the shorter time required for the preparation in Group C (2 h) than that required in Group B (at least 4 h) might have worked as effectively for the visibility of the distal ileum rather than the direct effect of colonoscopy. Although the optimal timing for bowel preparation has not yet been established, Choi CW et al. recommended that faster bowel preparation should be performed (within 6 h before SBCE) for improved distal small bowel examination [6]. Our results are considered to be consistent with this report.

Importantly, intestinal collapse tended to be observed more frequently in Group B. To determine lesions sensitively, the lumen should not be narrowed or collapsed. While the intestinal collapse that occurred more frequently in Group B might also have been due to the long time required for the ingestion of PEG, the precise mechanisms remain to be elucidated. Taking this into consideration, determining the timing of preparation in addition to determining the preparation methods before SBCE, is also a critical factor for the visibility of the intestinal mucosa during SBCE.

The PrepRICE study (ClinicalTrials.gov ID: NCT05140057) is currently underway in Portugal to investigate the optimal timing and bowel preparation by comparing four groups, each drinking a total of 1 L PEG at different times. The results of this study are awaited.

Our study has limitations. First, because this was a retrospective study, the compliance with bowel preparation used could not be verified. Second, this study did not examine the finding rates. A recent paper reported that there was no significant difference in finding rates between different timing of laxative intake, but there was a significant difference in visibility, especially in the distal small bowel [6]. Even if there was no significant difference in finding rates in our study, we believe it was useful.

In conclusion, despite the aforementioned limitations, our results suggest that taking 0.9 L of MC, which is easier to ingest than larger amounts of PEG, can be recommended as a standardized preparation method before SBCE, from the perspective of better visibility, especially that of the distal ileum, as well as the tolerability associated with the smaller volumes ingested. Further studies are required for greater optimization, such as the timing of preparation, to establish an optimal regimen for bowel preparation before SBCE.

## Figures and Tables

**Figure 1 diagnostics-13-03269-f001:**
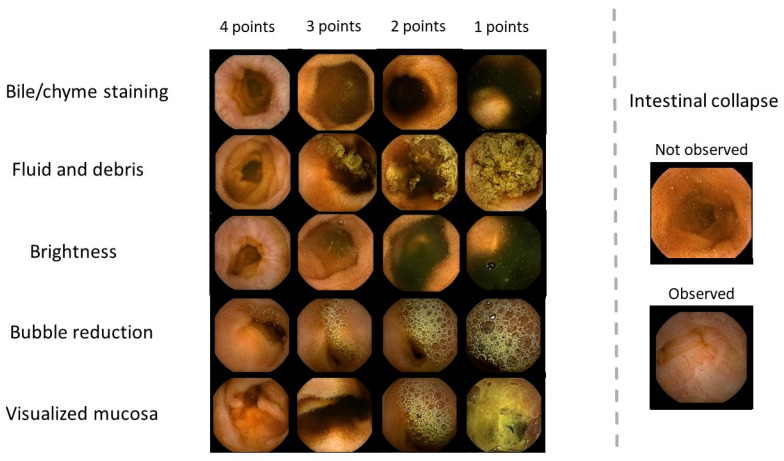
Scoring system images. Visibility was scored based on the levels of bile/chyme staining, fluid and debris, brightness, and bubbles. Each factor was scored from 1 (invisible) to 4 (visible) points. The status of intestinal collapse was determined as either observed (collapsed) or not observed (not collapsed).

**Figure 2 diagnostics-13-03269-f002:**
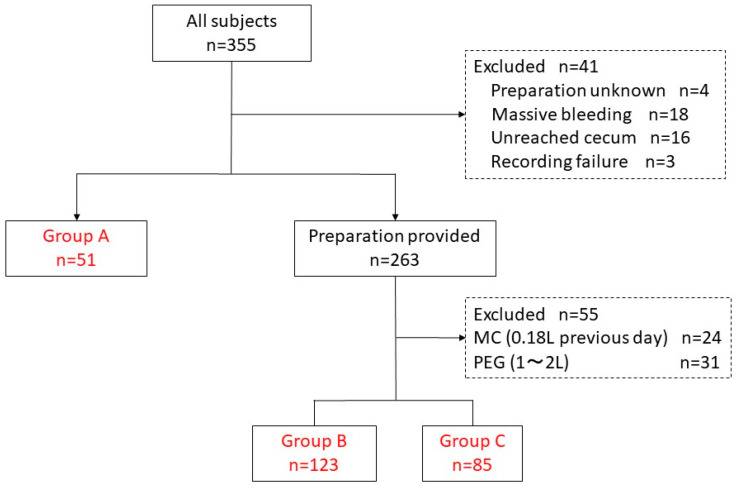
Flowchart of the cases enrolled in this study. Group A: overnight fasting only. Group B: bowel preparation with PEG (1–2 L) 4 h before colonoscopy and SBCE after overnight fasting. Group C: bowel preparation with MC 2 h before SBCE after overnight fasting.

**Figure 3 diagnostics-13-03269-f003:**
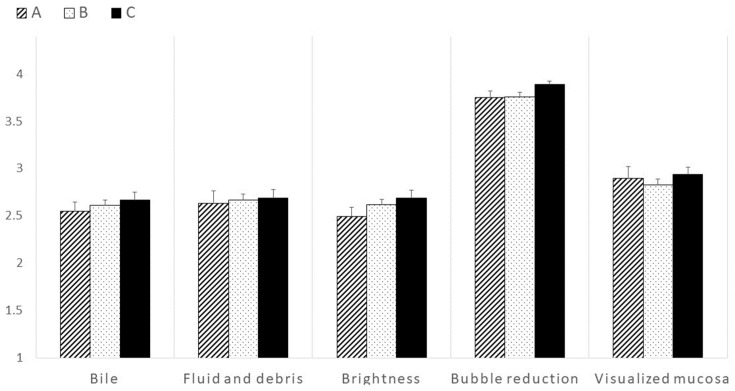
Comparison between the groups in terms of visibility based on each scoring parameter in the first 10 min to visualize the distal ileum. No significant differences were observed among the three groups in each scoring parameter.

**Figure 4 diagnostics-13-03269-f004:**
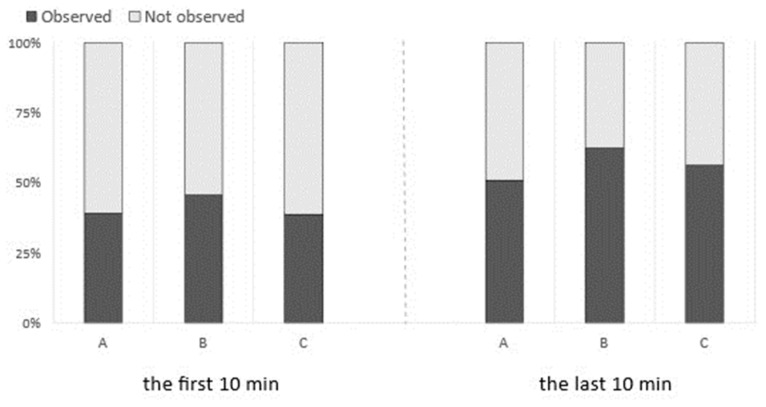
Assessment of the existence of intestinal collapse associated with the different preparation methods for SBCE. Although not statistically significant, Group B tended to reveal a collapsed intestine more frequently.

**Figure 5 diagnostics-13-03269-f005:**
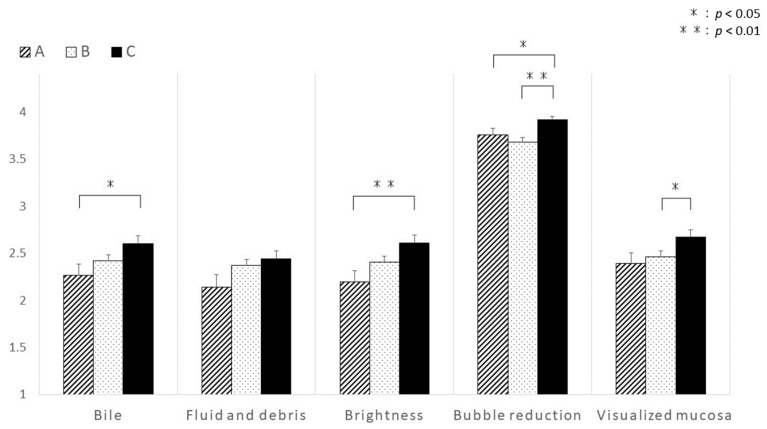
Comparison among the three groups in terms of the visibility based on each scoring parameter in the last 10 min to visualize the distal ileum. Group C revealed the best visibility based on the scoring parameters. The significance of differences between groups was determined via the Wilcoxon rank-sum test. *p*-values < 0.05 were considered statistically significant.

**Table 1 diagnostics-13-03269-t001:** Scoring system.

Bile/chyme staining
4 points: No bile
3 points: Some bile/chyme staining present, not interfering with observations
2 points: Quite a lot of bile/chyme staining, slightly hindering observations
1 point: Large amount of bile/chyme staining, hindering observations
Fluid and debris
4 points: No fluid and debris
3 points: Some fluid and debris present, not interfering with observations
2 points: Quite a lot of fluid and debris, slightly hindering observations
1 point: Large amount of fluid and debris, hindering observations
Brightness
4 points: Bright
3 points: Slightly dark, not interfering with observations
2 points: Quite dark, slightly hindering observations
1 point: Quite dark, hindering observations
Bubble reduction
4 points: Bubbles are observed of <20% on the image
3 points: Bubbles are observed of 20–49% on the image
2 points: Bubbles are observed of 50–79% on the image
1 point: Bubbles are observed of ≥80% on the image
Visualized mucosa
4 points: Visualization of ≥80% of mucosa
3 points: Visualization of 50–79% of mucosa
2 points: Visualization of 20–49% of mucosa
1 point: Visualization of <20% of mucosa
Intestinal collapse
Observed (collapsed) or not observed (not collapsed)

**Table 2 diagnostics-13-03269-t002:** Patient characteristic.

	Group A (No Preparation)	Group B (PEG)	Group C (MC)	*p* Value
Subjects number	51	123	85	
Age (years, mean ± SD)	63.7 ± 17.5	67.1 ± 18.0	63.2 ± 17.9	NS
Sex (male/female)	36/15	70/53	51/34	NS
Gastric transit time (min, mean ± SD)	54.2 ± 73.0	66.1 ± 62.9	55.4 ± 58.1	NS
Small bowel transit time (min, mean ± SD)	355.7 ± 182.9	296.6 ± 149.9	311.2 ± 138.9	NS
Indication of VCE n (%)				NS
OGIB	36 (70.6)	84 (68.3)	45 (52.9)	
Abdominal pain	4 (7.8)	5 (4.0)	6 (7.1)	
Abnormal bowel movement	1 (2.0)	6 (4.9)	9 (10.6)	
IBD (suspected)	5 (9.8)	8 (6.5)	16 (18.8)	
Tumor	3 (5.9)	14 (11.4)	7 (8.2)	
Others	2 (3.9)	6 (4.9)	2 (2.4)	

## Data Availability

Not applicable.

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
