# Peer review of "Optimal Bowel Preparation Method to Visualize the Distal Ileum via Small Bowel Capsule Endoscopy"

_diagnostics, 2023, doi:10.3390/diagnostics13203269_

Round 1

Reviewer 1 Report

I appreciated the idea of this study, as poor visibility of the terminal ileum still remains an issue, while many lesions are located in that area (e.g. Crohn’s disease). Even though the manuscript is well written (including tables and figures), with attention paid to details, I found two major problems, which preclude its publication.

1. Most important: In group B - Performing SBCE immediately after colonoscopy. Ileal intubation (with lots of water used in order to visualize the mucosa) could alter the results. Also, air insufflation may alter what really happens in the terminal ileum during SBCE. Therefore – MORE FLUIDS, MORE BUBBLES. I do not consider the group B adequate enough regarding accuracy in the interpretation of the data.

2. This one could be corrected. References are incredibly old. This issue renders “Introduction” and “Discussion” totally obsolete. Introduction: very old references [1-6] – most recent 2011, while a plethora of excellent manuscripts have been recently published (including 2023) and many regimens for bowel preparation have been used. Not including in the Background what is really known currently about a topic is unacceptable, as the current knowledge must be provided. Please revise and update. Moreover, the Authors inserted the ESGE Guidelines from 2018, while there has been an updated version published in 2022 (Pennazio et al). I strongly advise the Authors to insert recent data and resubmit their manuscript.

Still, the problem with the group B remains.

Other comments:

1. Title: Please delete “s” from Methods, as the manuscript refers to the optimal method.

2. Abstract:

a. The Authors wrote: “Group B (ingestion of 1–2 L polyethylene glycol before colonoscopy and SBCE after overnight fasting)”. Please mention that ingestion was 4 hours before colonoscopy and SBCE was performed immediately after colonoscopy.

b. Please insert the period the study was performed.

c. I suggest to present only 259 patients, as (from the main text) it appears that 96 were excluded. In fact, even the Authors wrote in line 136 “In total, 259 patients were enrolled in this study.”

3. Materials and Methods

Line 85: “Indigestion was performed” – please correct indigestion, as it does not make any sense, to “ingestion”.

4. Results are clearly presented, but based on wrong design. Therefore, they cannot be trusted.

5. Discussion

a. This paragraph is hampered by the incredibly old references. No proper discussion on the regimens for bowel preparation could be made without mentioning many other methods.

b. I suggest to start with their own results and then to make comments with reference to the recent scientific literature.

c. Limitations are only briefly discussed and not including all of them. Please be more generous.

d. Also, in order to increase the value of the manuscript, ongoing trials (from clinicaltrials.gov) should be inserted.

6. Conclusion is supported by the results. But, results are based on wrong design, therefore, they cannot be trusted.

Minor editing of the English language is required.

Reviewer 2 Report

Thank you for the opportunity to review the study titled: "Optimal Bowel Preparation Methods for Distal Ileum Visualization via Small Bowel Capsule Endoscopy." This retrospective study evaluates the visibility of the distal ileum in small bowel capsule endoscopy (SBCE) examinations with different prep protocol.

The research addresses a pertinent question given the absence of clear guidelines regarding necessary preparation methods. As endoscopists, we are keenly interested in optimizing the efficacy of SBCE for our patients.

In this study, the focus is primarily on visibility, utilizing a scoring system. It's essential to clarify whether the investigation also improved the rate of positive findings (which is the reason we send for SBCEs) between the different preparation groups, or if the evaluation solely pertains to visibility scores.

Moreover, it would be valuable to understand the criteria that determined patient assignment to different protocols. This could introduce potential bias stemming from varying patient characteristics, exam indications, and capsule systems. Incorporating details such as the year or period of the examination into the analysis could also be associated with different in protocols and different in visibility finding because different systems were use, which may affect visibility.

Abstract:

The number of patients mentioned in the abstract (355) should match the actual number of patients included in the study.

Methods:

Did the gastroenterologists responsible for scoring the exams underwent training or “calibration” to ensure standardization of the scoring system.

Given the retrospective nature of the study, more explanation of how patients were assigned to was group protocol is needed. While this aspect is partially addressed in the discussion, providing more comprehensive details is crucial. Many centers do not use SBCE immediately after a negative colonoscopy, so more information is needed.

Consider renaming the groups to reflect their respective protocols, making it easier for first-time readers to understand their significance.

Furthermore, What does “Indigestion was performed until no residual intestinal contents were observed. SBCE was performed immediately after the colonoscopy” mean as defining group B

Explain the selection criteria for the visibility score sub-components. Are these sub-components known to correlate with a higher yield of findings, or are they based on common sense? Providing a reference for the scoring system or subcomponent importance in finding rates would enhance clarity.

Consider comparing the finding rates among all patients with obscure gastrointestinal bleeding (OGIB) to determine if different preparation protocols yield varying efficacies. Findings is what we are all interested in, and showing a significant difference would make this paper more impactful.

Results:

Include percentages for categorical variables in the table

Provide insights into why the observed difference in preparation methods only appears to impact the terminal ileum.

Discuss potential limitations stemming from the retrospective nature of the study, such as variations in patient characteristics across different groups and the absence of data regarding specific findings.

Addressing these points will enhance the clarity and comprehensibility of the paper, making it more valuable to the readers and the medical community.

What does “Indigestion was performed until no residual intestinal contents were observed. SBCE was performed immediately after the colonoscopy” mean as defining group B

otherwise, Enlgish is sufficient.

Round 2

Reviewer 1 Report

Generally, the authors did their best to correct their paper, according to the reviewers’ suggestions/comments. Although I am not perfectly convinced about the group B being accurate, I consider that the new version of the manuscript explains in detail how things were performed and why.

Minor editing of English language is required.